# Tunneling between Multiple Histories as a Solution to the Information Loss Paradox

**DOI:** 10.3390/e25121663

**Published:** 2023-12-15

**Authors:** Pisin Chen, Misao Sasaki, Dong-han Yeom, Junggi Yoon

**Affiliations:** 1Leung Center for Cosmology and Particle Astrophysics, National Taiwan University, Taipei 10617, Taiwan; misao.sasaki@ipmu.jp (M.S.); innocent.yeom@gmail.com (D.-h.Y.); 2Department of Physics, National Taiwan University, Taipei 10617, Taiwan; 3Graduate Institute of Astrophysics, National Taiwan University, Taipei 10617, Taiwan; 4Kavli Institute for Particle Astrophysics and Cosmology, SLAC National Accelerator Laboratory, Stanford University, Stanford, CA 94305, USA; 5Kavli Institute for the Physics and Mathematics of the Universe (WPI), University of Tokyo, Chiba 277-8583, Japan; 6Center for Gravitational Physics, Yukawa Institute for Theoretical Physics, Kyoto University, Kyoto 606-8502, Japan; 7Department of Physics Education, Pusan National University, Busan 46241, Republic of Korea; 8Research Center for Dielectric and Advanced Matter Physics, Pusan National University, Busan 46241, Republic of Korea; 9Asia Pacific Center for Theoretical Physics, Pohang 37673, Republic of Korea; junggiyoon@gmail.com; 10Department of Physics, Pohang University of Science and Technology (POSTECH), Pohang 37673, Republic of Korea; 11School of Physics, Korea Institute for Advanced Study, Seoul 02455, Republic of Korea

**Keywords:** black hole, Hawking evaporation, information loss paradox, entanglement entropy, Euclidean path integral, Page time

## Abstract

The information loss paradox associated with black hole Hawking evaporation is an unresolved problem in modern theoretical physics. In a recent brief essay, we revisited the evolution of the black hole entanglement entropy via the Euclidean path integral (EPI) of the quantum state and allow for the branching of semi-classical histories along the Lorentzian evolution. We posited that there exist at least two histories that contribute to EPI, where one is an information-losing history, while the other is an information-preserving one. At early times, the former dominates EPI, while at the late times, the latter becomes dominant. By doing so, we recovered the essence of the Page curve, and thus, the unitarity, albeit with the turning point, i.e., the Page time, much shifted toward the late time. In this full-length paper, we fill in the details of our arguments and calculations to strengthen our notion. One implication of this modified Page curve is that the entropy bound may thus be violated. We comment on the similarity and difference between our approach and that of the replica wormholes and the islands’ conjectures.

## 1. Introduction

The information loss paradox of black holes [1] is an unresolved problem in modern theoretical physics. This paradox implies a contradiction between general relativity (GR) and local quantum field theory (QFT) [2,3]. There have been attempts to solve the paradox within GR and QFT. For example, the ‘soft hair’ proposed by Hawking, Perry. and Strominger [4] invokes the BMS symmetry within GR, but it was soon argued that the soft hair cannot carry information [5,6]. The ‘firewall’ conjecture [3], on the other hand, attempts to solve the paradox by violating the GR equivalence principle near the black hole horizon, but it was argued to be problematic [7,8]. More recently, there has been a new attempt to resolve this paradox by invoking the quantum gravitational corrections to the amplitude of Hawking radiation as a mechanism to release the information [9]. In order to render the black hole evaporation process unitary, it may be necessary to invoke some unknown new mechanism or a hidden sector [10,11,12] that lies outside proper GR and QFT. An implicit assumption is that such new elements must be deduced from *quantum gravity*.

It is important to stress that *entanglement entropy* is the physical quantity that measures the information flow from a black hole to its radiation [13]. In this bipartite system of a black hole and its Hawking radiation, the radiation entropy will increase as the evaporation proceeds. On the other hand, the black hole entropy, known as the Bekenstein entropy, which is supposed to describe the black hole’s microscopic degrees of freedom, will decrease as the black hole mass decreases. Then, at some point during the evaporation process, these two entropies must coincide with each other. Page asserted that the entanglement entropy of the system is approximately the minimum of the two [14]. This curve of the evolution of the entanglement entropy, known as the Page curve, with the turning point occurring at a time, the Page time, when the black hole area reduces to roughly half, plays an essential role in the investigation of the information loss problem.

In spite of Page’s demonstration in a quantum mechanical system, the attempt to derive the Page curve in GR has failed [15], which is often regarded as the deficiency of invoking the semi-classical perturbative calculations in GR. In particular, the decrease in the entanglement entropy after the Page time would require non-perturbative effects in quantum gravity beyond our current understanding of GR. However, a full-blown quantum gravity does not yet exist. One possible circumvention of a quantum gravity calculation would be via a new classical saddle point, e.g., a soliton, deduced from a valid approximation of quantum gravity, where a semi-classical tunneling around such a new saddle-point, e.g., via instantons, might be able to capture some essence of non-perturbative quantum effects evaluated around the original saddle-point. Under this light, the Page curve would result from a transition between the saddles.

Based on this philosophy, there exist two stages in black hole evaporation. First, before a modified Page time, GR and QFT work well and any hidden contributions are negligible. After the modified Page time, however, the hidden contributions are no more negligible. In fact, it must be dominant at the late times. Otherwise, the contributions via proper GR and QFT may erase the information.

Inspired by the above thinking, we have recently investigated [16] the information loss paradox via the Euclidean path integral (EPI) approach [17]. The EPI formalism is widely regarded as one of the most promising candidates of quantum gravity that can describe a non-perturbative processes [18]. Though not the final theory, EPI manages to capture the essence of a full-blown quantum gravity theory by dealing with the *entire* wave function, which includes not only perturbative but also non-perturbative gravity effects [19,20,21] via the Wheeler–DeWitt equation [22]. More specifically, we regard Hawking radiation as quantum tunneling and assume that, in addition to the perturbative channels that correspond to conventional Hawking radiation, there exist non-perturbative channels that turn the black hole geometry into trivial geometries, i.e., those without event horizons. We demonstrated in our short essay [16] that the essence of the Page argument remains valid, albeit with the Page curve modified where the Page time shifts significantly toward the late time of black hole evaporation. One implication of this modified Page curve is that the entropy bound may thus be violated. In this paper, we provide a more detailed and self-contained development of our arguments and calculations to support our claims.

This paper is organized as follows. In Section 2, we discuss the basics of the entanglement entropy and the physically essential conditions that explain the unitary Page curve. In Section 3, we discuss the EPI formalism and show that the previous essential conditions are indeed realized by EPI. In Section 4, we eventually obtain the modified form of the Page curve; we also argue that this formula is still consistent within the validity regime of the EPI. Finally, in Section 5, we summarize our results and comment on possible future applications.

## 2. Entanglement Entropy for Unitary Evaporation

### 2.1. Entanglement Entropy and the Page Curve

To quantitatively describe the flow of information, the notion of the *entanglement entropy* is found very useful [13]. Let us consider a system composed of two subsystems *A* and *B*, and a pure state given by |Ψ〉. The density matrix of the system is ρ≡|Ψ〉〈Ψ|. The reduced density matrix for subsystem *A* is given by tracing out subsystem *B*, i.e., $\rho_{A}\equiv {\mathrm{tr}}_{B}\rho$. Likewise, the reduced density matrix for subsystem *B* is given by $\rho_{B}\equiv {\mathrm{tr}}_{A}\rho$. Then, the von Neumann entropy of subsystem *A* is $S_{B}\left(A\right)\equiv -{\mathrm{tr}}_{A}\rho_{A}\ln\rho_{A}$. This is known as the entanglement entropy of subsystem *A*, which is the same as that of its complementary, $S_{A}\left(B\right)$, if the state is pure.

Let us consider quantum states of a black hole. We divide the system into a black hole, denoted by *A*, and Hawking radiation in the exterior, denoted by *B*. We assume that initially all degrees of freedom were in *A*, and, as time goes on, they are monotonically transmitted from *A* to *B* via Hawking radiation. According to the analysis by Page [14], by assuming a typical pure state with a fixed number of total degrees of freedom in the beginning, the entanglement entropy is almost the same as the Boltzmann entropy of the radiated particles. However, when the original entropy of the black hole decreased to approximately its half value, the entanglement entropy of the radiation begins to decrease (Figure 1). This turning time is called the *Page time*. If one further assumes that the Boltzmann entropy of the black hole is the same as the Bekenstein–Hawking entropy, one can compute the value of the Page time, which is approximately $∼M^{3}$ (in Planck units), where *M* is the black hole mass; it is evident that even at this time, the black hole remains semi-classical.

### 2.2. Wave Function of the Universe and Superposition of States

According to canonical quantum gravity [22], the entire information of the universe is included via the wave function of the universe Ψ, which is a function of the three-geometry $h_{\mu \nu }$ and a matter field configuration, say ϕ, on top of $h_{\mu \nu }$. This wave function should satisfy the quantum Hamiltonian constraint equation, the so-called Wheeler–DeWitt (WDW) equation. Since this is the fundamental equation of quantum gravity, unitarity must be manifest.

One can assume that the in-state of the wave function of the universe is given by $\left|\mathrm{in}\rangle \equiv \right|h_{\mu \nu }^{\left(\mathrm{in}\right)},\phi^{\left(\mathrm{in}\right)}\rangle$, where we assume that this in-state is a fixed classical configuration. This configuration will evolve to an out-state, say $\left|\mathrm{out}\rangle \equiv \right|h_{\mu \nu }^{\left(\mathrm{out}\right)},\phi^{\left(\mathrm{out}\right)}\rangle$. The WDW equation will determine the wave function of the universe for a given initial condition.

In order to understand the black hole evaporation process, we need to choose a proper out-state. It should be such that the observer at future infinity will see a semi-classical spacetime. However, the final out-state is not necessarily a unique classical spacetime; rather, it can be a *superposition of states corresponding to each classical spacetime* [23]. This follows from the observation that Hawking radiation can be interpreted as a result of quantum tunneling [21]. Thus
(1)$$\begin{array}{c} |\mathrm{out}\rangle =\sum_{i,\alpha }c_{\alpha ,i}|\alpha ;i\rangle , \end{array}$$
where |α;i〉 is a quantum state associated with a semi-classical state labeled by *i*, while α represents microscopic quantum degrees of freedom in the semi-classical state, and $c_{\alpha ,i}=\langle \alpha ;i|\mathrm{in}\rangle$ (Figure 2).

### 2.3. Essential Conditions toward the Unitary Page Curve

As mentioned in the above, a natural consequence from the picture based on canonical quantum gravity is that the out-state is a superposition of semi-classical states. Let us assume that one can categorize the semi-classical states into two distinct classes. The first class is those with *information-losing histories*, where the black hole keeps existing and loses information via Hawking radiation; therefore, the entanglement entropy monotonically increases up to the end point. The second class is those with *information-preserving histories*, which appear as a result of quantum tunneling, where there is no black hole, hence, no singularity nor event horizon. Hence, the entanglement entropy is zero for this class of histories.

When the in-state is dominated by information-losing histories, the (semi-classical) observer outside of the black hole cannot have access to the degrees of freedom inside of the black hole, which leads to the increase in the entanglement entropy. After the Page time, if the out-state is dominated by information-preserving histories, the observer can now have access to all degrees of freedom which would not have been measured in the information-losing histories. Hence, the entanglement entropy will vanish in the end. To summarize, in order to obtain a unitary Page curve, what one needs to justify is the following two essential conditions (Figure 3):–1. *Multi-history condition*: existence of multiple information-preserving and non-preserving histories;–2. *Late-time dominance condition*: dominance of the information-preserving history at the late time.
For simplicity, let us assume that there are only two histories, one information-losing and the other information-preserving. Then, one can approximately evaluate the entanglement entropy as $S_{\mathrm{ent}}≃p_{1}S_{1}+p_{2}S_{2}$, where 1 denotes the information-losing history, 2 denotes the information preserving-history, and $p_{i}$ and $S_{i}$ (i=1,2) are the probability and the entanglement entropy for each history, respectively. In the beginning, history 1 dominates and hence explains the increasing phase of the entanglement entropy. However, at the late times, history 2 dominates as $S_{2}=0$. The total entanglement entropy will eventually decrease to zero.

Therefore, if these two conditions are satisfied, then the unitary Page curve would be explained because the entanglement entropy eventually becomes zero again at the end. We will demonstrate that EPI can indeed deliver such a conclusion.

### 2.4. Entanglement Entropy with Multiple Histories

A generic quantum state with two classical histories can be described as follows: $|\psi \rangle =c_{1}|\psi_{1}\rangle +c_{2}|\psi_{2}\rangle$, where 1 and 2 denote the two different histories and $c_{1,2}$ are complex coefficients. The total density matrix ρ is
(2)$$\begin{array}{c} \rho =\left(\begin{array}{cc} |c_{1}{|}^{2} & c_{1}^{*}c_{2}\\ c_{1}c_{2}^{*} & |c_{2}{|}^{2} \end{array}\right). \end{array}$$We assume two histories are semi-classical and the off-diagonal terms will become less dominant; this is in accordance with the decoherence condition. With this assumption, one can write
(3)$$\begin{array}{c} \rho ≃\left(\begin{array}{cc} p_{1}\rho_{1} & 0\\ 0 & p_{2}\rho_{2} \end{array}\right)\left(I+e^{-S}\delta \right), \end{array}$$
where $p_{1}={\left|c_{1}\right|}^{2}$, $p_{2}={\left|c_{2}\right|}^{2}$, *I* is the identity matrix, and δ are the off-diagonal components that are approximately suppressed by $e^{-S}$, which is the transition amplitude between two histories. Although we already assumed the decoherence between two histories, we will see that this still approximately captures the physical essentials for the unitary evolution of the entire wave function.

Now let us assume that one can split the total system into two subsystems, *A* and *B*. Physically, *A* corresponds to the subsystem outside the horizon, while *B* is that inside the horizon (If there is no horizon, then *B* is empty.) By tracing out subsystem *B*, the reduced density matrix $\rho_{A}\equiv {\mathrm{tr}}_{B}\rho$ can be written as
(4)$$\begin{array}{c} \rho_{A}≃\left(\begin{array}{cc} p_{1}\rho_{1,A} & 0\\ 0 & p_{2}\rho_{2,A} \end{array}\right), \end{array}$$
where we can neglect the off-diagonal term δ due to the decoherence. This gives the entanglement entropy $S_{\mathrm{ent}}\left(A\right)$:(5)SentA=  −trρAlogρA(6)=   p1S1A+p2S2A−p1logp1−p2logp2,
where $S_{\nu }\left(A\right)\equiv -\mathrm{tr}\left(\rho_{\nu ,A}\log\rho_{\nu ,A}\right)$ is the entanglement entropy evaluated in each history (ν=1,2). Note that the last two terms are negligible if the number of degrees of freedom of the system is much greater than two. Therefore, we obtain the following approximate form of the entanglement entropy
(7)$$\begin{array}{c} S_{\mathrm{ent}}≃p_{1}S_{1}+p_{2}S_{2}, \end{array}$$
as advertised in the previous subsection.

## 3. Semi-Classical Euclidean Path Integrals

To confirm these essential conditions, one needs to compute the transition element 〈i|in〉, where |i〉 is a state representing a classical spacetime. Here we have omitted the label α for the microscopic degrees of freedom in each classical spacetime for notational simplicity. The probability of each history is given by $p_{i}={\left|\langle i|\mathrm{in}\rangle \right|}^{2}$. If we recover the label α, then we have $p_{i}=\sum_{\alpha }{\left|\langle \alpha ;i|\mathrm{in}\rangle \right|}^{2}$.

Although we do not yet have a final formulation for quantum gravity, at the semi-classical level, the Euclidean path integral can provide a good approximation that captures the essence of a bona fide full-blown quantum gravity theory [17]:(8)$$\begin{array}{c} \langle i|\mathrm{in}\rangle =\langle \left\{h_{\mu \nu }^{\left(i\right)},\phi^{\left(i\right)}\right\}|\left\{h_{\mu \nu }^{\left(\mathrm{in}\right)},\phi^{\left(\mathrm{in}\right)}\right\}\rangle =\int DgD\phi \;e^{-S_{E}\left[g_{\mu \nu },\phi \right]}, \end{array}$$
where $S_{E}$ is the Euclidean action and the integral is over all configurations of four geometries $g_{\mu \nu }$ and matter fields ϕ that match those on the two different space-like surfaces $|\{h_{\mu \nu }^{\left(\mathrm{in}\right)},\phi^{\left(\mathrm{in}\right)}\}\rangle$ and $|\{h_{\mu \nu }^{\left(i\right)},\phi^{\left(i\right)}\}\rangle$. This path integral can be well approximated by summing over on-shell solutions, i.e., either Lorentzian classical solutions or Euclidean instantons.

### 3.1. Hawking Radiation as Instantons

We consider the following model:(9)$$\begin{array}{c} S_{E}=-\int \sqrt{+g}d^{4}x\left(\frac{R}{16\pi }-\frac{1}{2}{\left(\partial \phi \right)}^{2}\right)-\int_{\partial M}\frac{K-K_{o}}{8\pi }\sqrt{+h}d^{3}x, \end{array}$$
where R is the Ricci scalar, ϕ is a massless scalar field, and K and $K_{o}$ are the Gibbons–Hawking boundary terms at infinity of the solution and of the Minkowski background, respectively.

Now we consider the path integral from an in-state that includes a black hole to a possible out-state. If we approximate the in-state as a pure Schwarzschild geometry, then the most typical instanton that connects the past infinity and the future infinity will be the Euclidean Schwarzschild geometry. Figure 4 shows this geometry conceptually. For a given constant *t*, the hypersurface is analytically continued to the Euclidean manifold. The past (lower) part has no scalar field and this mimics a vacuum in-state, while the future (upper) part has a non-trivial scalar field configuration. Also, note that a non-trivial scalar field configuration must satisfy the equation of motion on top of the Euclidean–Lorentzian manifold configuration.

Let us see further details of scalar field perturbations near the horizon. As we magnify around the Einstein–Rosen bridge, we can approximate a solution as a superposition of in-going and out-going modes [21]. In order to satisfy the classicality (i.e., reality) condition at future infinity, we must choose a condition that there are only real-valued out-going modes at (III). This implies that the solution must be complex valued in (I) and (II). In addition, since the out-going scalar field carries energy δM, the green-colored region surrounding the event horizon has mass $M^{\prime }=M-\delta M$. By choosing the Euclidean time period $\tau_{T}=8\pi M$ to cancel out the boundary term at infinity, the horizon becomes a cusp singularity. Nevertheless, it can be shown that the cusp does not cause an issue as it can be appropriately regularized with the result independent of the regularization [24].

In [21], it was argued that free scalar field perturbations on the Euclidean–Lorentzian manifold can be identified as Hawking radiation. This is justified from computing the probability of such configurations. For each on-shell history, the tunneling rate is given by $\Gamma \propto e^{-B}$, where
(10)$$\begin{array}{c} B=S_{E}\left(\mathrm{instanton}\right)-S_{E}\left(\mathrm{background}\right). \end{array}$$After regularizing the cusp [24], we obtain
(11)$$\begin{array}{c} B=4\pi \left(M^{2}-M^{\prime 2}\right), \end{array}$$
where *M* and $M^{\prime }$ are the mass of the initial black hole and that of the final black hole, respectively (We note that $S_{E}\left(\mathrm{instanton}\right)$ is defined as that for the solution with half of the period in [21] so that it represents the amplitude of the tunneling wave function. Here, in accordance with the standard convention, we define the Euclidean action to be the one with the whole period.)

For a large black hole with M≫1 and ω≪M, we have
(12)$$\begin{array}{c} B=8\pi M\omega \;. \end{array}$$This is perfectly consistent with Hawking radiation, where the Hawking temperature is $T_{H}={\left(8\pi M\right)}^{-1}$. In the other extreme, if we choose δM=M, the black hole should transit to a Minkowski background. In general, there exists such a process toward a trivial geometry, as long as one assumes a massless scalar field. The only price to pay is that the probability will be exponentially suppressed for δM=M≫1.

### 3.2. Thin-Shell Toy Model for Tunneling to Trivial Geometry

The existence of the instanton that describes a transition process from a black hole to a flat space is evident from the discussion of the previous subsection. However, in order to obtain the solution faithfully, we need to solve the equations for time-dependent instantons with metric back reactions fully taken into account. This means we need to solve the highly nonlinear, coupled partial differential equations, which is beyond the scope of the present paper. Nevertheless, we expect that the result is not qualitatively different from the one extrapolated from the case of δM≪M.

To support our expectation, as a simple toy model for the tunneling of a black hole to a flat space, we consider a thin-shell model [25]. Namely, we mimic Hawking radiation using a thin-shell nucleated in the black hole geometry. After nucleation, we assume that it carries all the energy as it moves away to future null infinity. Hence, the spacetime inside the shell will be flat, while outside the shell will be the Schwarzschild geometry with the mass equal to that of the black hole prior to the tunneling [21].

To be specific, we consider a spherically symmetric spacetime with the metric,
(13)$$\begin{array}{c} ds_{\pm }^{2}=-f_{\pm }\left(R\right)dT^{2}+f_{\pm }^{-1}\left(R\right)dR^{2}+R^{2}d\Omega^{2}\;, \end{array}$$
with a thin-shell located at R=r, whose intrinsic metric is given by
(14)$$\begin{array}{c} ds^{2}=-dt^{2}+r^{2}\left(t\right)d\Omega^{2}, \end{array}$$
where the region outside (inside) the shell R>r (R<r) is denoted by + (−). We impose the metric ansatz for outside and inside the shell, $f_{\pm }\left(R\right)=1-2M_{\pm }/R$, where $M_{+}$ and $M_{-}$ are the mass parameters of each region. We assume $M_{+}>M_{-}=0$; hence, there is no black hole inside the shell.

The equation of motion of the thin shell is determined via the junction equation [25]:(15)$$\begin{array}{c} \epsilon_{-}\sqrt{{\dot{r}}^{2}+f_{-}\left(r\right)}-\epsilon_{+}\sqrt{{\dot{r}}^{2}+f_{+}\left(r\right)}=4\pi r\sigma \left(r\right)\;, \end{array}$$
where we impose $\epsilon_{\pm }=+1$ so that the outward normal vector of the shell has the proper expansion (i.e., the extrinsic curvature). The above equation can be re-expressed as
(16)$$\begin{array}{c} {\dot{r}}^{2}+V_{\mathrm{eff}}\left(r\right)=0, \end{array}$$
where
(17)$$\begin{array}{c} V_{\mathrm{eff}}\left(r\right)=f_{+}-\frac{{\left(f_{-}-f_{+}-16\pi^{2}\sigma^{2}r^{2}\right)}^{2}}{64\pi^{2}\sigma^{2}r^{2}}. \end{array}$$Here, σ(r) is the tension of the shell, which satisfies the energy conservation equation,
(18)$$\begin{array}{c} \frac{d\sigma }{dr}=-\frac{2\sigma \left(1+w\right)}{r}\;, \end{array}$$
where *w* is the equation of the state of the shell, and we require w≥−1 to satisfy the null energy condition. In general, the tension may be assumed to have the form [26],
(19)$$\begin{array}{c} \sigma \left(r\right)=\sum_{i=1}^{n}\frac{\sigma_{i}}{r^{2(1+w_{i})}}\;, \end{array}$$
where *n* is an arbitrary positive integer and $\sigma_{i}$ and $w_{i}$ are constants.

Now we look for a static solution. In the Euclidean time, a static instanton may be regarded as a thermal instanton. The condition for the existence of such a solution is the existence of a radius $r_{0}$ such that $V_{\mathrm{eff}}\left(r_{0}\right)=V_{\mathrm{eff}}^{\prime }\left(r_{0}\right)=0$. This is satisfied, for example, if one chooses $w_{1}=0.5$, $w_{2}=1$, and appropriately tunes $\sigma_{1}$ and $\sigma_{2}$, as depicted in Figure 5.

Using the static shell instanton, we may now compute the tunneling probability of the black hole geometry to the one with no black hole as shown in Figure 6, where an evaporating black hole geometry (left panel) tunnels to the geometry of an expanding shell with no black hole (right panel) at the hypersurface denoted by *t*. In the language of the previous section, the left panel corresponds to the information-losing history, $h_{1}$, and the right to the information-preserving history, $h_{2}$ (see Figure 3). Under normal circumstances the information-preserving geometry is exponentially suppressed. However, we will show that it becomes dominant at the late times.

### 3.3. Tunneling Probability

The Euclidean action is given by
(20)$$\begin{array}{c} S_{E}=-\int \frac{R}{16\pi }\sqrt{+g}d^{4}x+\sigma \int_{\Sigma }\sqrt{+h}d^{3}x-\int_{\partial M}\frac{K-K_{o}}{8\pi }\sqrt{+h}d^{3}x\;, \end{array}$$
where R is the Ricci scalar, Σ is the hypersurface of the thin shell, and K and $K_{o}$ are the Gibbons–Hawking boundary terms at infinity for the solution and the Minkowski background, respectively [27].

Now we evaluate $S_{E}$ for the static thin-shell instanton with the Euclidean time period $\Delta T_{E}=8\pi M_{+}$. By using the on-shell condition, we can simplify it as
(21)$$\begin{array}{c} S_{E}\left(\mathrm{thin}-\mathrm{shell}\right)=\left(\frac{\sigma }{2}+\lambda \right)\int_{\Sigma }\sqrt{+h}d^{3}x+\left(\mathrm{boundary}\;\mathrm{term}\right), \end{array}$$
where λ is the pressure of the shell. From the energy conservation relation, we have
(22)$$\begin{array}{c} \lambda \left(r_{0}\right)=-\left(\sigma \left(r_{0}\right)+\frac{r_{0}}{2}\sigma^{\prime }\left(r_{0}\right)\right)\;. \end{array}$$In addition, assuming $f_{-}\left(r_{0}\right)=1$ and $f_{+}\left(r_{0}\right)=1-2M_{+}/r_{0}$, we can derive an important relation by taking the *r*-derivative of (Equation 15),
(23)$$\begin{array}{c} M_{+}=-\sqrt{f_{+}}4\pi r_{0}^{2}\left(\sigma \left(r_{0}\right)+r_{0}\sigma^{\prime }\left(r_{0}\right)\right). \end{array}$$Noting that Equation (Equation 22) implies $-\left(\sigma /2+\lambda \right)=\left(\sigma +r\sigma^{\prime }\right)/2$ for $r=r_{0}$, and using the above relation, we find
(24)$$\begin{array}{c} S_{E}\left(\mathrm{thin}-\mathrm{shell}\right)=4\pi M_{+}^{2}+\left(\mathrm{boundary}\;\mathrm{term}\right)\;. \end{array}$$Since the action $S_{E}\left(\mathrm{background}\right)$ of the Euclidean Schwarzschild metric is purely given by its boundary term at infinity, and it coincides with that of $S_{E}\left(\mathrm{thin}-\mathrm{shell}\right)$, we obtain
(25)$$\begin{array}{c} B=S_{E}\left(\mathrm{thin}-\mathrm{shell}\right)-S_{E}\left(\mathrm{background}\right)=4\pi M_{+}^{2}\;. \end{array}$$We note that this agrees with the extrapolation of Equation (Equation 11) to the case of $M^{\prime }=M-\delta M=0$.

Here, we mention a point that has an important implication to our later discussion. In general, there is no freedom in the choice of the Euclidean time period of the background (the Schwarzschild metric in our case); it is fixed. On the other hand, the Euclidean time period of the solution is not unique, but can be multiples of the fundamental period determined via the background geometry. Therefore, we should take into account all possible multiple period instanton configurations. Then, for an instanton with *n* periods, we obtain
(26)$$\begin{array}{ccc} B_{n} & = & n\left[(\mathrm{bulk}\;\mathrm{action})+(\mathrm{boundary}\;\mathrm{term})\right]-\left(\mathrm{boundary}\;\mathrm{term}\right)\\ & = & (2n-1)S\;, \end{array}$$
where $S=4\pi M_{+}^{2}$ and n≥1. The total tunneling probability is therefore
(27)$$\begin{array}{c} \Gamma =\sum_{n=1}^{\infty }e^{-S(2n-1)}=\frac{1}{e^{S}-e^{-S}}\;. \end{array}$$Evidently, n=1 is the dominant contribution and one recovers the result $\Gamma =e^{-S}$ for S≫1. However, the subdominant contributions may become important for small *S*. This has an important consequence when we consider the Page curve.

For simplicity, we consider only two classes of histories, where one is the semi-classical black hole which gradually loses its mass due to Hawking radiation (hence a class composed of a single history), and the other is a family of spacetimes with the thin shell emitted to infinity as a result of the tunneling. It may be noted that, since the tunneling can happen at any time, histories will continue to branch out. Eventually, infinitely many histories of the second class appear in the out-state (Figure 3) [23,26].

Let us denote the probability of the semi-classical black hole history by $p_{1}$ and that of the information-preserving histories by $p_{2}$. From Equation (Equation 27), they are estimated as
(28)$$\begin{array}{c} p_{1}=\frac{e^{S}-e^{-S}}{1+e^{S}-e^{-S}}\;,\;p_{2}=\frac{1}{1+e^{S}-e^{-S}}\;. \end{array}$$Interestingly, $p_{2}$ becomes greater than $p_{1}$ at S≃1 (Figure 7). We note that this is true even if one ignores the effect of the multiple Euclidean time-period instantons, though the value of *S* at which $p_{2}$ exceeds $p_{1}$ would be somewhat different.

## 4. Modified Page Curve

### 4.1. Page Curve

Now we are ready to evaluate the Page curve. Using the formula for the expectation value of the entanglement entropy (Equation 7), and noting that the entanglement entropy vanishes for the information-preserving histories, we obtain
(29)$$\begin{array}{ccc} S_{\mathrm{ent}} & = & \left(S_{0}-S\right)\times p_{1}+0\times p_{2}\\ & = & \left(S_{0}-S\right)\left(\frac{e^{S}-e^{-S}}{1+e^{S}-e^{-S}}\right), \end{array}$$
where $S_{0}$ is the initial entropy of the black hole; $S_{0}=4\pi M_{0}^{2}$, where $M_{0}$ is the initial mass. Here we have assumed that the entanglement entropy for the semi-classical black hole monotonically increases as its mass decreases due to Hawking radiation. Namely, the entropy of the emitted radiation $S_{\mathrm{rad}}=S_{0}-S$ exactly accounts for the entanglement entropy. The entanglement entropy as a function of $S_{\mathrm{rad}}$ is depicted in Figure 8. It is clear that the entanglement entropy increases monotonically, reaches a maximum, and eventually vanishes as the black hole mass decreases to zero. This result is consistent with the notion of unitary evolution.

We have thus successfully recovered the most important property of the Page curve, i.e., the preservation of unitarity, albeit with the price that it is significantly modified. In particular, an important and interesting observation is that the turning point, i.e., the Page time, is far beyond the moment when the Bekenstein–Hawking entropy decreased to its half value. In our picture, it occurs when $S∼\log S_{0}$, instead of $S∼S_{0}/2$ for the conventional Page time. This implies that the equivalence of the Bekenstein–Hawking areal entropy and the Boltzmann entropy is violated. However, one should realize that this equivalence is not a result of any fundamental principles. In fact, an intriguing counter example has recently been pointed out in [28], which is perfectly consistent with both local field theory and semi-classical general relativity.

### 4.2. Validity of the Semi-Classical Approximation

Throughout this paper, we have assumed the validity of the semi-classical approximation for quantum gravity. It is therefore important to check if it remains valid in the regime of our interest. The crucial point is whether the modified Page curve we obtained would be within the validity of our semi-classical approximation. This can be checked by inspecting the maximum of the curve. By taking the derivative of $S_{\mathrm{ent}}$ with respect to *S*, we obtain the modified Page time at which the entanglement entropy is maximum, i.e., the time at which $dS_{\mathrm{ent}}/dS=0$ is satisfied. This gives an implicit equation for the value of *S* at the maximum as a function of $S_{0}$,
(30)$$\begin{array}{c} S_{0}=S_{m}+\left(1+2\sinh S_{m}\right)\tanh S_{m}, \end{array}$$
where $S_{m}$ is the areal entropy at the Page time (Figure 9). For $S_{0}\gg 1$, one approximately obtains $S_{m}≃\log S_{0}$.

This result implies that as long as the initial black hole entropy is sufficiently large ($S_{0}\gg 1$), the areal entropy at the Page time can be sufficiently greater than the Planck scale ($\log S_{0}\gg 1$). Therefore, the turning point of the Page curve occurs while the semi-classical approximation is still perfectly valid, provided that the initial black hole is macroscopic (For example, even for a black hole of a fairly small mass $M_{0}∼10^{5}$g, $S_{0}∼10^{20}$; hence, $S_{m}∼46$.)

## 5. Conclusions

### 5.1. Recent Developments Based on String Theory

It is interesting to compare our approach with the recent developments based on string theory, where the Page curve was semi-classically reproduced by evaluating quantum extremal surfaces (QES). It was found [29,30], in two-dimensional gravity, that a saddle of QES without an *island* is dominant before the Page time, which induces the increase in the entanglement entropy. After the Page time, the QES is dominated by the other saddle, which has an island, and this saddle results in the decrease in the entanglement entropy. It was also argued in [31] that such a role played by the island can also reproduce the Page curve in higher-dimensional black holes.

The string-based islands approach and ours share two essential features. One, there exist more than two contributions to the evolution of the entanglement entropy, where one is dominant at the early time and the other at the late time. Two, the final entanglement entropy is dominated by the late-time condition. That is, the multi-history condition and the late-time dominance condition in our Euclidean path integral approach are analogous to the spirit of the QES computations for black holes.

However, the physical interpretation of the islands’ conjecture is still not very clear. One reason is that the entanglement entropy computed in QES is based on the density matrix instead of the quantum state, which is the standard quantum field theory approach and what we have followed. It is therefore natural to ask, what is the implication of the islands or the replica wormholes [32,33] to the more orthodox, state-level path integral in Lorentzian signatures, and vice versa?

Although our interpretation of the thin-shell tunneling shares common features with the islands’ conjecture and the replica wormholes, there exists a clear difference. In spite of significant progress in the islands’ conjecture, it is still premature to understand the classical geometry of the new saddle from QES. For example, the island is an extreme of the generalized entropy, rather than the original path integral itself. Moreover, the replica wormhole is based on the Euclidean path integral, whereas our approach deals with the branching of semi-classical histories along the Lorentzian evolution. In this sense, our interpretation of thin-shell tunneling may be more closely connected with the recent studies in the context of real-time gravitational replica wormholes [34,35,36] for the generalization of the islands’ conjecture and the replica wormholes with baby universes [37,38,39]. In [21], it was argued that Hawking radiation can be viewed as instanton. Likewise, the instanton interpretation of Hawking radiation after the Page time might provide an alternative means to understand black hole information paradox. More detailed comparisons of our approach with the replica wormholes approach, either Euclidean or Lorentzian, are left for future investigations.

### 5.2. Connection with Black Hole Remnants

It has been argued that, based on the generalized uncertainty principle (GUP), the black hole Hawking evaporation would end with Planck-size black hole remnants (BHR) [40]. There have been suggestions that BHR holds the secret of solving the information loss paradox [41,42,43] (For an overview of black hole remnant as a solution to the information loss paradox, see [43].) In this paper, we follow the conventional view to assume that a black hole would evaporate completely at the end. It would be interesting to investigate the impact of BHR to our scenario of the black hole entropy evolution based on the instanton tunneling between multiple histories.

### 5.3. Future Prospects

We have argued that the canonical quantum gravity with the Euclidean path integral approach can provide a consistent picture to resolve the information loss paradox. By computing the wave function of the universe with the Euclidean path integral, we successfully justified the two essential conditions, that is, the *multi-history condition* and the *late-time dominance condition*, and eventually obtained a modified Page curve that preserves the unitarity, but with the Page time shifted significantly towards the late time.

Note that the entanglement entropy of a black hole can never exceed its Boltzmann entropy. Therefore, if one insists on the equivalence between the Bekenstein–Hawking areal entropy and the Boltzmann entropy, then the entanglement entropy cannot exceed the Bekenstein–Hawking areal entropy. On the contrary, one salient outcome of our computation is that there exists a moment where the entanglement entropy is greater than the Bekenstein–Hawking areal entropy. This necessarily implies that the number of states inside the horizon must have been accumulated during the black hole evaporation, although such an accumulation is strictly bounded. We emphasize that this violation of the equivalence is not in contradiction with basic principles of physics [28].

In our picture, the turning point of the Page curve, though shifted significantly towards the end life of the black hole evaporation, is still in the semi-classical regime of quantum gravity as we have shown. Hence, there might be a way to experimentally investigate our notion. If our model can be examined not only via theoretical means, but also via experimental methods [44], then the synergy between theory and experiment may hopefully lead us to the ultimate understanding of the information loss paradox.

## Figures and Tables

**Figure 1 entropy-25-01663-f001:**
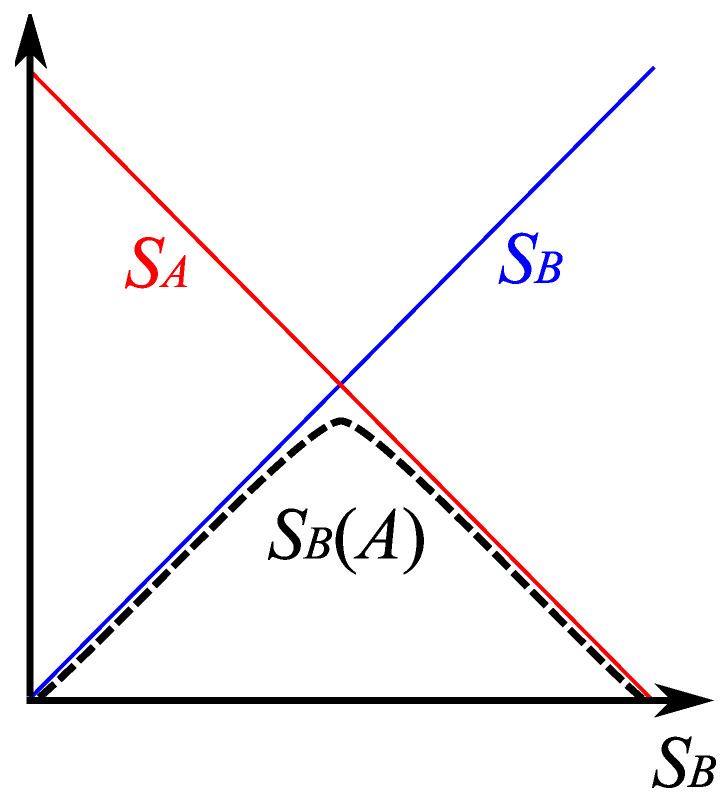
The Page curve, where $S_{A}$ and $S_{B}$ are Boltzmann entropy of *A* and *B*, respectively, and $S_{B}\left(A\right)$ denotes the entanglement entropy.

**Figure 2 entropy-25-01663-f002:**
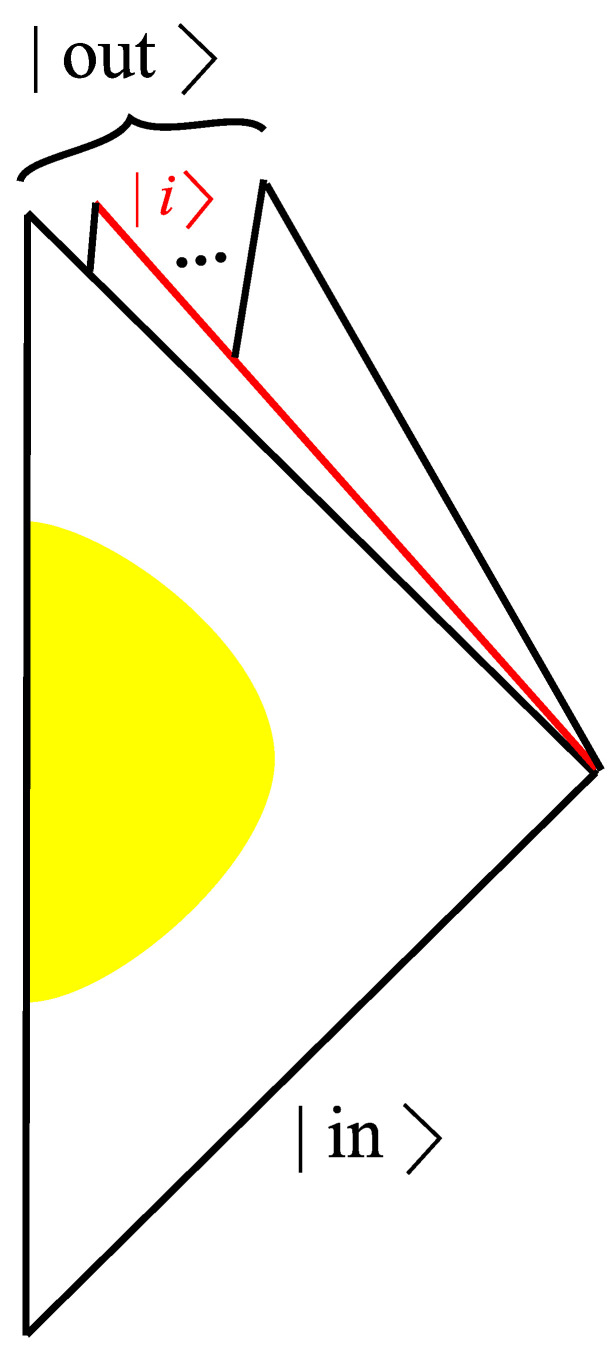
The path integral from the in-state |in〉 to the out-state |out〉, where the out-state is a superposition of classical boundaries {|i〉}.

**Figure 3 entropy-25-01663-f003:**
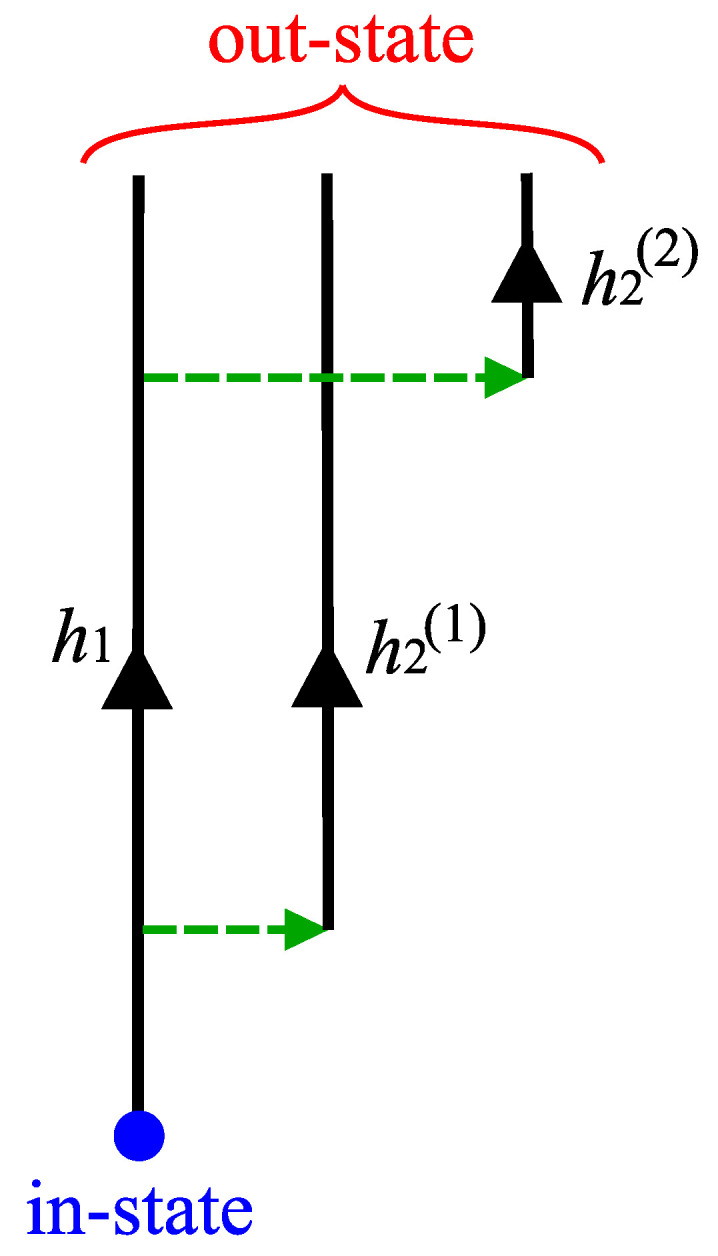
Conceptual figure for interpretations. $h_{1}$ is the information-losing history, while $h_{2}^{\left(1,2\right)}$ are the information-preserving history. For any time, histories can be branched; the tunneling probability must be dominated at the late time.

**Figure 4 entropy-25-01663-f004:**
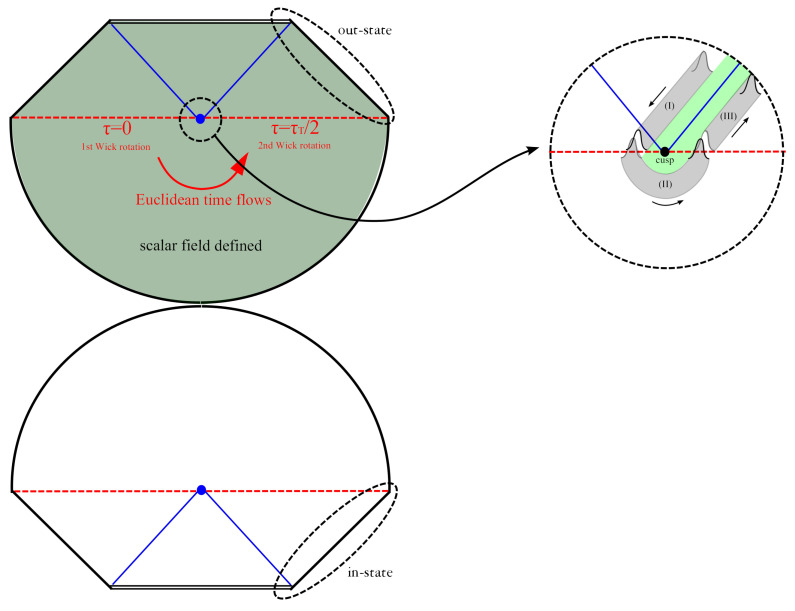
Typical instantons for Hawking radiation [21].

**Figure 5 entropy-25-01663-f005:**
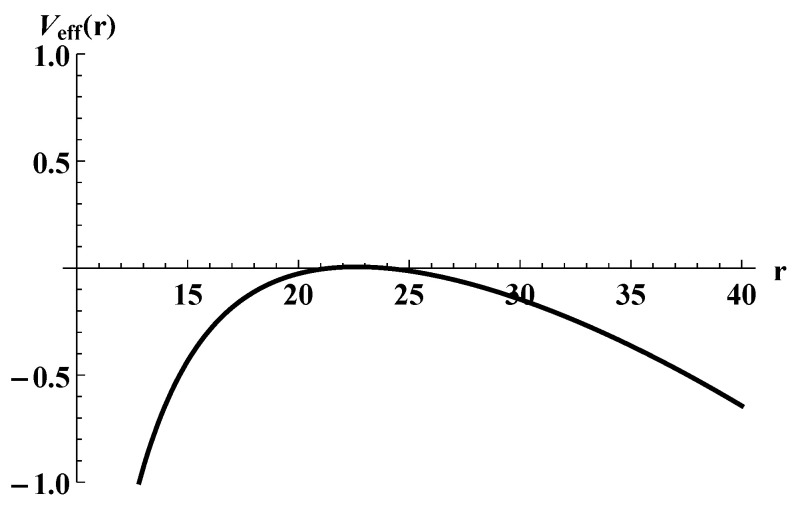
The effective potential $V_{\mathrm{eff}}\left(r\right)$ that allows a static shell solution. The parameters are $w_{1}=0.5$, $\sigma_{1}≃340$ and $w_{2}=1$, $\sigma_{2}=10$. The black hole mass is set to $M_{+}=10$.

**Figure 6 entropy-25-01663-f006:**
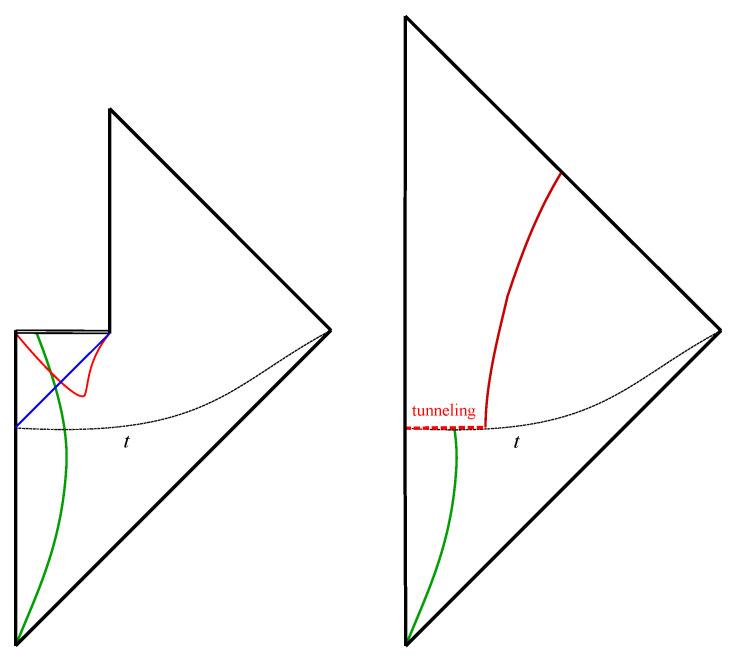
(**Left**): the causal structure of the usual semi-classical black hole, where the green curve is the trajectory of the collapsing matter, the red curve is the apparent horizon, and the blue line is the event horizon. (**Right**): the causal structure after a quantum tunneling at the time slice *t*. After the tunneling, matter or information (red curve) is emitted and the black hole structure disappears.

**Figure 7 entropy-25-01663-f007:**
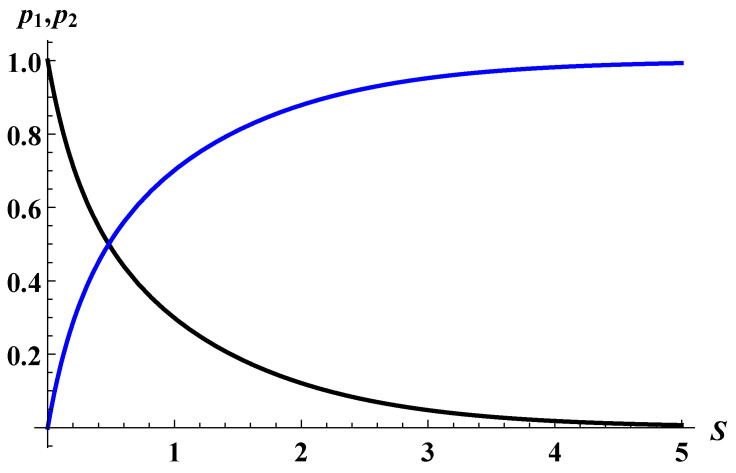
Probabilities of the semi-classical history ($p_{1}$, blue curve) and histories with thermal-shell emission ($p_{2}$, black curve) as a function of the entropy *S*. For large *S*, $p_{1}$ is dominated; however, there exists a golden cross between two probabilities, and eventually, $p_{2}$ is dominated.

**Figure 8 entropy-25-01663-f008:**
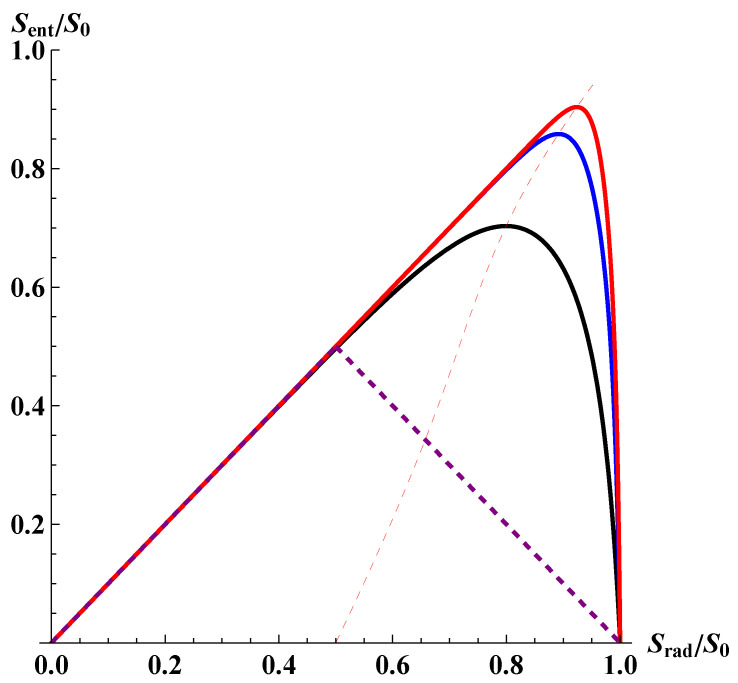
Entanglement entropy, $S_{\mathrm{ent}}/S_{0}$, vs. entropy of the emitted radiation, $S_{\mathrm{rad}}/S_{0}\equiv 1-S/S_{0}$. As a demonstration, we display curves with different initial black hole entropies (therefore different initial masses) $S_{0}=10$ (black), 30 (blue), 50 (red), respectively, to illustrate the tendency. The purple dashed curve is the conventional Page curve. Note that our modified Page curve deviates from the conventional Page curve. The thin red dashed curve is the location of the modified Page time, i.e., $dS_{\mathrm{ent}}/dS_{\mathrm{rad}}=0$. The more massive the initial black hole, the more skewed the modified Page curve, with the modified Page time shifted more significantly towards the late time.

**Figure 9 entropy-25-01663-f009:**
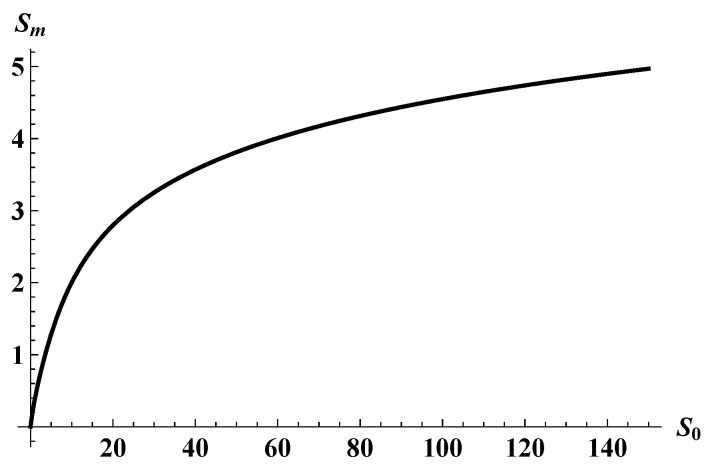
The maximum entanglement entropy of the modified Page curve, $S_{m}$, vs. the initial areal entropy $S_{0}$.

## Data Availability

Data is contained within the article.

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
