# Peer review of "Tunneling between Multiple Histories as a Solution to the Information Loss Paradox"

_entropy, 2023, doi:10.3390/e25121663_

Round 1

Reviewer 1 Report

Comments and Suggestions for Authors

Manuscript: Entropy-2682427

Dear Editor,

I have read the article entitled "Tunneling between Multiple Histories as a Solution to the Information Loss Paradox," authored by Chen et al and submitted for publication in your esteemed journal. The paper explores a thought-provoking solution to the information loss paradox associated with black hole Hawking evaporation, shedding light on the challenging problem of reconciling quantum mechanics with general relativity.

The abstract of the paper provides a clear overview of the paper's content, highlighting the authors' approach to resolving the information loss paradox through a combination of Euclidean path integral (EPI) formalism and the concept of branching semi-classical histories. The motivation behind their research, which seeks to address the limitations of perturbative calculations in general relativity (GR) and quantum field theory (QFT), is well articulated. In particular, the authors propose that non-perturbative quantum effects could be captured by considering a new classical saddle point derived from an approximation of quantum gravity. This novel perspective is intriguing and has the potential to offer valuable insights into the behavior of black holes and the preservation of information during their evaporation.

The organization of the paper is in general clear and logical. The authors provide a structured framework, with sections dedicated to explaining the entanglement entropy, the EPI formalism, the modified Page curve, and a summary of their findings. The presentation of the modified Page curve, along with its implications for the entropy bound, is a significant contribution to the field of black hole physics and quantum gravity.

I would like to make a suggestion for the authors to consider in their paper. In the concluding remarks of their article, the authors argue that their model suggests a turning point of the Page curve, albeit significantly shifted towards the late-time stages of black hole evaporation. This shift, according to their analysis, still places the phenomenon within the semi-classical regime of quantum gravity. While the authors have made a compelling case for their approach, I believe that it is essential to consider the potential implications of remnants in the context of their model. Remnants, as remnants of black holes, have been a subject of interest and debate in the study of black hole evaporation. There are theoretical discussions in the literature about the possible existence of remnants, even after complete Hawking radiation, as suggested by some former references:

Phys. Lett. B 678 (2009) 131.

International Journal of Theoretical Physics 50 (10): 3212-3224, 2011

JHEP 1008:089,2010; Erratum-ibid.1101:021,2011

Astrophys. Space Sci. 340, 155 (2012)

The authors should acknowledge this possibility in their article and address how their model accommodates or contrasts with the concept of remnants. Including a discussion on the implications of remnants, if their model permits, would enhance the completeness and robustness of their argument. I recommend that the authors revise their paper to include a section or discussion that addresses the potential existence of remnants in the context of their model. They should consider the arguments and findings from relevant previous research that discuss remnants, and explore whether their model supports or contradicts these ideas. By addressing this aspect, the authors can provide a more comprehensive analysis of their proposed solution to the information loss paradox.

I look forward to reviewing the revised version of the article, and I believe that this additional discussion will strengthen the overall quality and impact of the work. Please convey my appreciation to the authors for their valuable contribution to the field, and I look forward to the opportunity to review the revised manuscript.

Sincerely.

Author Response

Response to the first reviewer’s comments:

We appreciate very much the valuable comments made by the reviewer about the implications of black hole remnants (BHR) to the information loss paradox, in particular to our scenario of instanton tunneling between multiple histories in the black hole evolution. We follow the suggestion and created a new subsection entitled “Connection with Black Hole Remnants” in the Conclusion Section. In addition, we checked through the papers that the Reviewer suggested to us and have identified two of them that are more relevant to the issue. In addition, we add two additional citations, one relates to the argument for the existence of Planck-size BHR, which is also cited by several of the references suggested by the Reviewer, and the other is a paper that provides an overview of the BHR and the information loss paradox.  

We sincerely wish that this revision is acceptable.

Reviewer 2 Report

Comments and Suggestions for Authors

This paper further explores the ideas first explained in their essay to the gravity research foundation that was awarded a honorable mention

Resolving information loss paradox with Euclidean path integral

Indeed, by introducing a branching of histories, the authors obtain a more reasonable behaviour that can accomodate for the Page phenomenon,

they introduce very interesting effects, like the . Wave Function of the Universe consisting of a Superposition of States. All appears to me very interesting and consistent, so I certainly recommend publication

Author Response

Thank you for your appreciation of our work. 

Round 2

Reviewer 1 Report

Comments and Suggestions for Authors

Dear Editor,

The authors have diligently addressed the concerns and suggestions raised during the initial review process, significantly improving the clarity, coherence, and overall quality of the manuscript. Their incorporation of refined explanations and strengthened arguments has notably enhanced the article's contribution to the field. The proposed solution regarding tunneling between multiple histories to address the Information Loss Paradox presents a thought-provoking and innovative perspective. I believe this work will serve as a valuable contribution to the existing literature on this complex and important topic. I commend the authors for their dedication and commendable efforts in revising the manuscript. I am confident that this article will be of great interest and benefit to the readers of Entropy. 

* There is a flaw about one of the references, which appeared as "?" at line 402:  paradox [40? ,41]. (For an overview of black hole remnant as a solution to the information